# Sugar-Sweetened Beverages, Foods of Low Nutritional Value, and Child Undernutrition in Cambodia

**DOI:** 10.3390/ijerph21020169

**Published:** 2024-02-01

**Authors:** Akemi Brown, Margaret Trimble, Karen Sokal-Gutierrez, Lia Fernald, Kristine Madsen, Bathsheba Turton

**Affiliations:** 1University of California San Francisco Joint Medical Program, University of California, Berkeley, CA 94720, USA; ksokalg@berkeley.edu; 2School of Public Health, University of California, Berkeley, CA 94720, USA; margarettrimble@berkeley.edu (M.T.); fernald@berkeley.edu (L.F.); madsenk@berkeley.edu (K.M.); 3University of Puthisastra, Phnom Penh 12211, Cambodia; bturton@bu.edu

**Keywords:** stunting, wasting, malnutrition, Cambodia, sugar-sweetened beverages, ultra-processed foods

## Abstract

Child undernutrition persists in Cambodia despite recent progress. As Cambodia undergoes a shift in dietary consumption that coincides with economic, demographic, and epidemiologic changes, there is risk of ultra-processed foods and sugar-sweetened beverages displacing nutrient-dense foods during the critical period of infant growth in the first 24 months. The aim of this study was to assess the introduction and intake of foods of low nutritional value and sugar-sweetened beverages and their association with undernutrition among children 24 months of age in rural and semi-urban Cambodia. Cross-sectional analyses of a 24-h dietary recall from a sample (*n* = 377) of 24-month-olds found that the majority of infants had been introduced to packaged salty snacks and sweets by 12 months of age and to sugar-sweetened beverages by 15 months. By 24 months of age, 78% of children had consumed foods of low nutritional value and 57% consumed a sugar-sweetened beverage on the previous day. Multivariate logistic regression analyses demonstrated that infant intake of a flavored sugary drink on the previous day was associated with over two times the odds of both stunting and wasting, and consumption of packaged sweets on the previous day was associated with over two times the odds of wasting, but no association was found with stunting. These findings underscore the need to improve educational and policy interventions to support healthy feeding practices for infants and young children.

## 1. Introduction

Globally, 22% of children under the age of five suffer from stunting (low height-for-age) and 6.7% from wasting (low weight-for-height), with higher proportions in low-income countries [1]. Child undernutrition continues to represent a significant global public health problem that contributes to an estimated 45% of child deaths and has significant adverse implications for children’s physical, cognitive, and behavioral development during childhood and later in life [2].

Cambodia, a rapidly developing lower-middle income country, has made considerable improvements in child nutrition over the past two decades [3]. However, approximately 22% of children under the age of five years are still stunted and 10% are still wasted [3]. Furthermore, infant and young child feeding practices (IYCF) in Cambodia are inadequate. Among children 6–23 months, only 42% met the minimum acceptable diet according to the World Health Organization (WHO) IYCF guidelines [3,4]. Since the first 1000 days from conception through age two are a critical period to reduce stunting, improving IYCF practices during this time is essential [5]. Specifically, the WHO recommends exclusive breastfeeding in the first 6 months of life and introduction of nutritionally adequate complementary foods from 6–23 months [4].

Like many low- and middle-income countries, Cambodia’s rapid economic growth has been accompanied by a transition away from traditional dietary patterns towards westernized diets, including ultra-processed foods with added sugars, fats, and refined carbohydrates and sugar-sweetened beverages (SSBs) [6]. Rising consumption of these unhealthy foods and beverages by Cambodian children has been documented in the most recent Cambodia Demographic and Health Survey, which found that among children 6–23 months, 21% consumed unhealthy foods, 28% consumed sweet beverages, and 29% consumed no vegetables or fruits on the previous day [3]. High consumption of unhealthy snack foods and beverages has been associated with lower nutrient intake and stunting among children in LMICs [7,8]. One explanation for this is that unhealthy foods and beverages may displace nutrient-dense foods that are critical during the first two years of life [7,9]. Other studies have demonstrated that increased consumption of ultra-processed foods has been associated with a higher prevalence of overweight/obesity and cardiometabolic comorbidities among children [10] and that high SSB consumption is associated with an increased risk for type 2 diabetes [11]. Given the persistence of undernutrition in Cambodia and rising rates of child obesity throughout Asia [12], consumption of unhealthy foods and sugar-sweetened beverages may have important implications for the concurrent development of undernutrition and obesity in Cambodia [13,14,15].

There is a need for further investigation of the health impacts of infants’ consumption of foods of low nutritional value, such as ultra-processed foods and sugar-sweetened beverages, particularly in rural and semi-urban Cambodia, where the majority of these infants reside. While there is a growing body of nutrition research in Cambodia, few studies have explicitly examined the introduction and frequency of consumption of different categories of low nutritional value foods and beverages and the association with malnutrition [16]. The aim of this study was to assess the consumption of foods of low nutritional value and sugar-sweetened beverages, as well as the association of intake of these foods and beverages with undernutrition among children 24 months of age in rural and semi-urban Cambodia.

## 2. Materials and Methods

This is a cross-sectional, secondary analysis of data collected through Cambodia SMILE, a pilot study that aimed to determine the effectiveness of a community-based health promotion program to reduce dental caries in children from infancy to 24 months of age. It included an intervention arm that received an oral health intervention (fluoride application, oral health education) and a control arm [17]. The protocol was approved by the National Ethics Committee for Health Research (NECHR #0306) under the Ministry of Health in Cambodia.

Cambodia SMILE recruited child-caregiver dyads from community health centers in rural and semi-urban districts of Kampong Speu Province. Rural and semi-urban districts were selected as this is where the majority of infant-aged children live; additionally, as the study was conducted in conjunction with the Ministry of Health and Cambodian government, the study districts were selected by these government stakeholders. Data were collected from 2015 to 2017 by trained primary healthcare providers at child ages of 9, 12, 15, 18, and 24 months using research examination forms (Appendix A). After informed consent was obtained, caregivers were interviewed in the Khmer language by trained native-speaking interviewers. Basic sociodemographic data including parent occupation, income (reported in Cambodian riels then converted to United States dollars (USD)), parent age, parent education, and household characteristics were collected. At each study visit, caregivers were asked if they had introduced various foods and beverages, and if so, the age (in months) at which the child was first exposed to the following list: dietary staples (rice, meat, vegetables, fruit, egg); foods of low nutritional value (FLNV) (packaged salty snacks, such as crackers; packaged sweets, such as candy and cookies; and Khmer sweet cake—a traditional Cambodian rice cake that is typically made of glutinous rice flour dough and palm sugar); low- or no-sugar added beverages (water, breastmilk, bottled milk formula); and SSBs (flavored sugary drinks without nutrient content which included fruit juices, sodas, and syrups; and sugar-sweetened drinks with some nutrient content, including sweetened condensed milk, sweetened soy milk, and sweetened rice water). Caregivers also reported the frequency of consumption by the child on the previous day of selected foods at specific time points (morning, middle of the day, afternoon, evening, and night); the methodology was comparable to a food frequency questionnaire isolated to the last 24 h. The food list for the questionnaire was adapted from a prior study by Turton et al. [18] focused on foods and beverages associated with the development of dental caries among children including those most commonly consumed. A “packaged food” variable was created by combining packaged salty snacks and packaged sweets. The study population for this secondary analysis consists of a sub-sample of 377 children and their caregivers and uses data collected at the 24-month follow-up only.

Trained researchers assessed each child’s length/height and weight at the 24-month visit, including recumbent length in children under 2 years of age or unable to stand up and height of children aged 2 years and older; weight was measured with a calibrated precise scale. Children were measured twice and the mean value was used according to WHO guidelines [19]. The following anthropometrics were calculated according to WHO Child Growth Standards [19]: length/height-for-age z-score/height-for-age z-score (LAZ/HAZ), weight-for-age z-score (WAZ), BMI-for-age z-score (BAZ), and weight-for-height z-score (WHZ). Child malnutrition status was categorized as stunted (LAZ/HAZ < −2 SD below median), wasted (WHZ < −2 SD below median), and underweight (WAZ < −2 SD below median).

The control and intervention data from the original Cambodia SMILE study were merged to increase the sample size for the secondary analysis. A sensitivity analysis was conducted to confirm the robustness of the findings by analyzing the original control and intervention groups separately using the same multivariate models (Appendix A). Cases with relevant missing data were excluded from the logistic regressions.

Data were entered into, cleaned, and analyzed in STATA 15 SE. Tables for the sensitivity analysis and figures were created in R (version 4.3.0). Descriptive analyses assessed the sociodemographic and nutritional characteristics of the study population, as well as the introduction and previous-day consumption of the selected foods and beverages. Bivariate associations of FLNV and SSB—based on any consumption (binary) as well as increasing levels of consumption (ordinal)—were examined. Within the SSB category, only the subcategory of flavored sugary drinks met our criteria for inclusion in multivariate analysis (i.e., bivariate *p*-value < 0.20). We then performed staged multivariable models. An initial multivariable model examined the associations of stunting and wasting with sociodemographic variables, with *p* < 0.20 resulting in a final base model including child sex, household income, and maternal occupation, as well as a variable indicating subject intervention in the original Cambodia SMILE study. Statistical significance was considered for *p*-values < 0.05.

## 3. Results

### 3.1. Sociodemographic and Nutritional Status Characteristics

Children had a mean age of 22.7 months, with approximately equal representation of sexes (Table 1). Most families were low-income, with a monthly household income less than $250 (average GDP per capita in Cambodia for the period of data collection—2015–2017—was $1284 [20]). Most parents had a primary school level of education or less, and rural and semi-urban households were nearly equally represented.

Many children experienced undernutrition, with 43% of the infants being stunted, 16% being wasted, and 27% being underweight.

### 3.2. First Introduction of Foods and Beverages

By 3 months of age, very few children had been introduced to the specified foods and beverages, except milk formula and water (Figure 1). Roughly a quarter of caregivers reported introducing their child to milk formula and slightly less than one-third reported introducing their child to water.

By 6 months of age, a larger proportion of infants had been introduced to a wider variety of foods and beverages. A substantial proportion of infants (11–23%) had been introduced to foods of low nutritional value, including packaged sweets, packaged salty snacks, and Khmer sweet cake, and to flavored sugary drinks.

By 12 months of age, a greater proportion of children had been introduced to all selected foods and beverages. Over half of children had been introduced to packaged sweets and Khmer sweet cake and two-thirds had been introduced to packaged salty snacks. Likewise, nearly half of children had been introduced to flavored sugary drinks.

By 24 months of age, the majority of children had been introduced to FLNV and SSB. Three-quarters of children had been introduced to packaged sweets and Khmer sweet cake, while almost all had been introduced to packaged salty snacks. Finally, almost two-thirds of children had been introduced to flavored sugary drinks.

### 3.3. Frequency of Reported Food and Beverage Intake at 24 Months

At 24 months of age, reported consumption of selected foods on the previous day shows that most children had consumed rice or rice porridge (96%), meat (79%), and vegetables (65%) (Figure 2). In addition, over three-quarters (78%) of children had consumed at least one FLNV, making it the third most frequently consumed food category; and over one-quarter had consumed an FLNV three or more times on the previous day. The most commonly consumed FLNV were packaged salty snacks (59%), followed by packaged sweets (33%) and Khmer sweet cake (28%).

At 24 months of age, reported consumption of selected beverages on the previous day shows that the majority (92%) drank water, and approximately one-quarter were breastfed (27%) and given milk formula (22%) (Figure 3). Furthermore, over half (57%) of children had consumed at least one SSB, and one-fifth had consumed an SSB three or more times on the previous day. The most commonly consumed SSBs were flavored sugary drinks (24%) and sweetened condensed milk (23%).

### 3.4. Association between Food and Beverage Intake and Stunting at 24 Months of Age

At 24 months, consumption of a flavored sugary drink on the previous day was significantly associated with higher odds of stunting in the adjusted model (OR 2.02; 95% CI [1.18, 3.43]) (Table 2). This association was driven by the control group after sensitivity analysis (Appendix A). There was a trend towards decreased odds of stunting with increased levels of consumption of any FLNV or any packaged salty snacks on the previous day. This trend appears to be primarily driven by the intervention subgroup in which consumption of higher levels of FLNV or packaged salty snacks was significantly associated with decreased odds of stunting (Appendix A).

### 3.5. Association between Food and Beverage Intake and Wasting at 24 Months of Age

At 24 months, consumption of a flavored sugary drink on the previous day was significantly associated with greater odds of wasting in adjusted models (OR 2.21; 95% CI [1.20, 4.08]) (Table 3). Higher odds of wasting was associated with consumption of two FLNV on the previous day (i.e., Khmer sweet cake, packaged salty snack, or packaged sweet) compared to no FLNV consumption on the previous day (OR 2.64; 95% CI [1.10, 6.33]). This finding appeared to be primarily driven by the intervention subgroup (Appendix A). In addition, higher odds of wasting was associated with consumption of one packaged sweet on the previous day compared to no consumption of packaged sweets on the previous day (OR 2.22; 95% CI [1.11, 4.44]). The directionality of these relationships observed in the overall cohort were maintained in the sensitivity analysis (Appendix A).

## 4. Discussion

In this study sample from rural and semi-urban Cambodia, a substantial proportion of children had been introduced to sugar-sweetened beverages and/or ultra-processed snacks in the first 6 months of life, contrary to WHO IYCF guidelines [4]. Cumulative introduction to these unhealthy foods and beverages increased steadily with age. The majority of children had been introduced to packaged salty snacks and packaged sweets by 12 months and to sugar-sweetened beverages by 15 months. By 24 months, three-quarters of children consumed foods of low nutritional value on the previous day. Compared to nutrient-dense foods, foods of low nutritional value at this age were consumed at higher frequency than vegetables and fruits and almost equal frequency to meat. Additionally, at 24 months, one-half of children consumed a sugar-sweetened beverage on the previous day.

Multivariate logistic regression analysis demonstrated that prior-day child intake of a flavored sugary drink was significantly associated with over two times the odds of both stunting and wasting, and prior-day consumption of one packaged sweet was associated with over two times the odds of wasting. The association between increased odds of stunting and prior day consumption of a flavored sugary drink was largely driven by the control subgroup (Appendix A). This may be due to a social desirability bias resulting in under reporting of flavored sugary drink intake among the intervention subgroup participants who received education on the relationship between sugar intake and dental caries.

This study also found a trend towards decreased stunting with increased number of FLNV and any packaged salty snack consumed on the prior day. This protective association may be related to the specific types and quantity of packaged salty snacks consumed, which our study did not examine in detail. We hypothesize that perhaps children who ate more packaged salty snacks—that had been purchased—were from families with higher income, which might have afforded access to more nutritious foods for the mothers during pregnancy as well as their young children; and therefore, this association may be confounded by income in a way that we were unable to demonstrate in this study. Prior studies in India have demonstrated similar findings with lower rates of stunting and wasting with higher rates of junk food consumption (Athavale et al., 2020) [21]. Sensitivity analysis demonstrated that this trend was largely driven by the intervention subgroup and, moreover, that within the intervention subgroup the opposite directionality was seen with the consumption of these foods and increased odds of wasting. Since stunting is a measure of chronic malnutrition and wasting is a measure of acute malnutrition, this opposing directionality may reflect recency differences in terms of when children were consuming these foods that this study was not designed to capture.

The infant and young child feeding practices in this sample are contrary to the World Health Organization and UNICEF recommendations to exclusively breastfeed infants for the first 6 months, and from 6–23 months to introduce nutritionally adequate complementary solid foods including daily fruits and vegetables and to avoid ultra-processed foods and sugar-sweetened drinks [4]. Our finding that most children frequently consumed foods of low nutritional value and sugary drinks indicates that these unhealthy products have become a staple of infant diets in rural and semi-urban Cambodia. These findings are consistent with previous studies in Cambodia and other low- and middle-income countries [14,22,23,24]. Data collection for this study was performed between 2015–2017, yet remains consistent with more recent analysis from Hinnouho et al. [25] demonstrating high rates of commercial food consumption among infants and with the most recent Cambodia Demographic and Health Survey [3].

Though the prevalence of undernutrition among infants and children has declined significantly in Cambodia since the time of data collection for this study (2015–2017), Cambodia continues to have unacceptably high rates of stunting and wasting [3]. At 24 months of age, the association between prior-day consumption of sugar-sweetened beverages with significantly increased odds of stunting and wasting is likely explained by evidence that non-nutritious diets reported at 24 months reflect a longstanding pattern of non-nutritious diets at earlier infant ages [25]; and, over the first 2 years of life, the cumulative impact of sugary drinks and non-nutritious snack foods displacing nutrient-dense foods and beverages can lead to stunting and wasting [26,27]. Factors that contribute to this displacement include the increasing marketing and availability of low-cost and convenient processed foods and beverages and the subsequent development of taste preferences for these food products among young children [21,28,29,30,31,32]. Children with high sugar intake have been found to consume less protein, total fat, and key vitamins compared to those with low sugar intake [33]. Other studies have found that high consumption of ultra-processed foods and sugar-sweetened beverages corresponded to low dietary diversity [34], which in turn has been associated with stunting [35,36], and greater dietary diversity has been associated with higher HAZ scores [37,38].

The association between high sugar intake and undernutrition may also be mediated by dental caries. Early and frequent consumption of sugary drinks and snacks is known to cause dental caries [39], and dental caries have been shown to be associated with increased risk of stunting and wasting in low- and middle-income countries, including Cambodia [40,41,42,43]. The hypothesized mechanisms include chronic oral infection/inflammation, oral pain, difficulty eating and sleeping, and disruption of the hypothalamic–pituitary–adrenal axis [44,45,46].

In all, these findings of unhealthy children’s diets and associations with stunting and wasting are particularly concerning as similar dietary shifts throughout Asia have contributed to a double burden of malnutrition—with undernutrition persisting alongside obesity at the individual, household, and population level—placing children at risk of poor growth and development as well as non-communicable diseases (NCDs) such as diabetes and cardiovascular disease [47,48,49,50,51]. Displacement of nutrient-dense foods due to excessive intake of saturated fats and consumption of foods high in added sugars may play a role in measures of overnutrition [52] as well as undernutrition. It is therefore critical to develop effective measures to improve infant diets, including programs for caregiver and child education on healthy food and beverage choices, as well as policies to limit the sale of sugary drinks and ultra-processed food products, and to provide families with better access to affordable nutrient-rich food and beverage options.

Limitations of this study include purposive sampling and small sample size, which limits the generalizability of the findings, and its cross-sectional design, which precludes the ability to make causal inferences. Survey responses may be affected by recall and social desirability bias. While sensitivity analysis confirmed the directionality of the observed associations, the potential for unmeasured confounding remains. Additionally, the dietary questionnaire was not developed according to WHO Infant and Young Child Feeding (IYCF) guidelines, nor was it validated against other previously validated dietary survey instruments [53,54], limiting further detailed nutritional analysis. The strengths of this study include its contribution to the literature regarding the introduction and frequency of consumption of packaged foods and sugar-sweetened beverages in addition to nutrient-dense foods in children’s diets and the significant associations between consumption of sugary beverages and foods of low nutritional value with increased odds of stunting and wasting. Future studies should explore other variables such as food and beverage composition and portion size using a culturally appropriate validated survey instrument and maternal health characteristics such as BMI, tobacco use, and intergenerational malnutrition resulting from famine during the Khmer Rouge era. In addition, mediation analysis could be utilized to capture more nuanced effects of diet—especially the unique role of sugar—on anthropometric measures, particularly stunting, and to elucidate mediating mechanisms such as nutrient displacement and dental caries.

## 5. Conclusions

This study found that children in rural and semi-urban Cambodia were introduced to foods of low nutritional value and sugar-sweetened beverages at very young ages, and these products have become a routine part of the infant and young child diet, in contrast to IYCF guidelines. Furthermore, consumption of these products was significantly associated with over two-fold odds of stunting and wasting. While more detailed investigation of the consumption of specific products remains to be conducted, more effective educational and policy interventions are needed to promote healthy IYCF practices and reduce consumption of sugary beverages and commercially packaged foods of low nutritional value.

## Figures and Tables

**Figure 1 ijerph-21-00169-f001:**
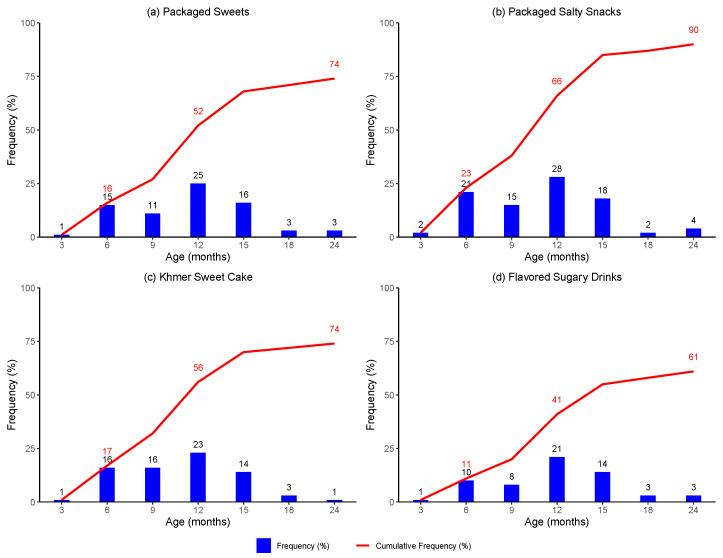
Introduction to selected food and beverage products by age in months.

**Figure 2 ijerph-21-00169-f002:**
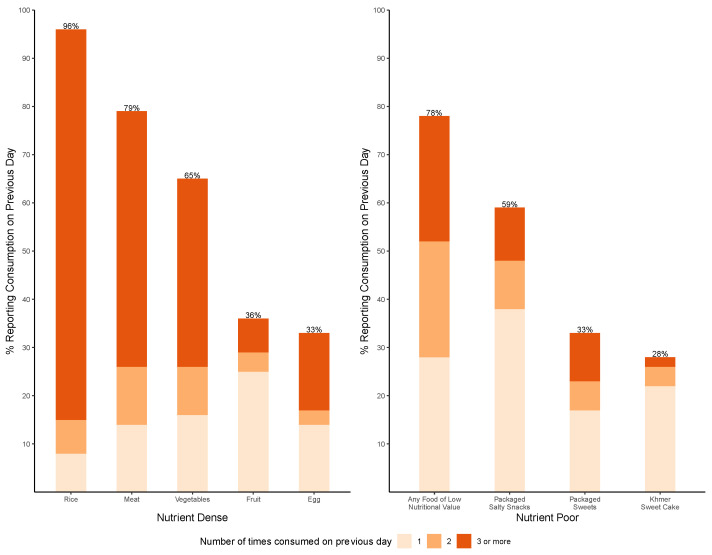
Frequency of consumption of reported foods on previous day at 24 months of age.

**Figure 3 ijerph-21-00169-f003:**
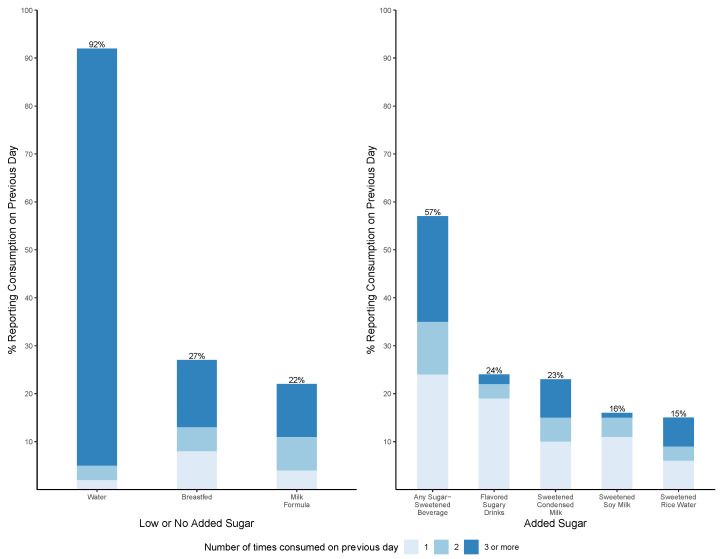
Frequency of consumption of reported beverages on previous day at 24 months of age.

**Table 1 ijerph-21-00169-t001:** Sociodemographic, health, and nutritional status characteristics of study population.

Characteristics	Mean ± SD or N (%), n = 377
Child	
Sex	
Male	169 (44.8)
Female	208 (55.2)
Age (months)	22.7 ± 2.9
Household Characteristics	
Monthly income (USD ^†^)	
Less than $50	29 (8.1)
$51–150	87 (24.2)
$151–250	130 (36.2)
More than $250	113 (31.5)
Electricity	320 (84.7)
Running water	103 (27.3)
Number of people in household	5.0 ± 1.7
Cooking fuel source	
Wood	241 (63.9)
Charcoal	16 (4.2)
Gas	50 (13.3)
Combination	70 (18.6)
Location	
Rural	167 (45.0)
Semi-urban	204 (55.0)
Parent Characteristics	
Mother’s age (years)	28.1 ± 5.5
Father’s age (years)	30.1 ± 5.9
Parent education	
Unable to read	49 (13.1)
Primary school	173 (46.3)
High school	141 (37.7)
University	11 (2.9)
Mother’s occupation	
Stay-at-home	98 (26.0)
Seller	19 (5.0)
Farmer	22 (5.8)
Government worker	3 (0.8)
Other	235 (62.3)
Father’s occupation	
Stay-at-home	34 (9.1)
Seller	19 (5.1)
Farmer	39 (10.4)
Government worker	12 (3.2)
Other	271 (72.3)
Nutritional Status of Child	
Stunted (LAZ ^‡^/HAZ ^‡^ < −2 SD below median)	156 (43.1)
Wasted (WAZ ^‡^ < −2 SD below median)	58 (15.7)

^†^ USD, United States dollars; ^‡^ LAZ, length-for-age z-score; HAZ, height-for-age z-score; WAZ, weight-for-age z-score.

**Table 2 ijerph-21-00169-t002:** Association between flavored sugary drinks and foods of low nutritional value consumed on previous day and stunting at 24 months ^†^.

Exposure Variable	OR	[95% CI]	*p*-Value
Any flavored sugary drinks ^‡^	2.02	[1.18–3.43]	0.01
Any FLNV ^‡^	0.89	[0.51–1.55]	0.69
Number of FLNV on previous day			
One	1.01	[0.54–1.91]	0.96
Two	0.94	[0.47–1.86]	0.86
Three or more	0.68	[0.34–1.34]	0.27
Any packaged salty snacks	0.69	[0.43–1.09]	0.11
Number of packaged salty snacks on previous day			
One	0.74	[0.44–1.23]	0.24
Two or more	0.58	[0.31–1.11]	0.10
Any packaged sweets	1.03	[0.63–1.68]	0.91
Number of packaged sweets on previous day			
One	1.24	[0.67–2.32]	0.49
Two or more	0.82	[0.42–1.61]	0.57

^†^ Odds ratios were generated using multivariate logistic regression; separate models adjusted for child sex, household income, mother’s occupation, and subject assignment in the original SMILE study were run for each exposure variable; N = 333–335 (cases with missing relevant data were excluded from analysis); ^‡^ flavored sugary drinks include fruit juice, sodas, and syrups; Foods of Low Nutritional Value (FLNV) include packaged salty snacks, packaged sweets, and Khmer sweet cake.

**Table 3 ijerph-21-00169-t003:** Association between flavored sugary drinks and foods of low nutritional value consumed on the previous day and wasting at 24 months ^†^.

Exposure Variable	OR	[95% CI]	*p*-Value
Any flavored sugary drinks ^‡^	2.21	[1.20–4.08]	0.01
Any FLNV ^‡^	1.79	[0.84–3.79]	0.13
Number of FLNV on previous day			
One	1.81	[0.78–4.20]	0.17
Two	2.64	[1.10–6.33]	0.03
Three or more	1.12	[0.43–2.91]	0.82
Any packaged salty snacks	1.34	[0.73–2.44]	0.34
Number of packaged salty snacks on previous day			
One	1.48	[0.78–2.82]	0.23
Two or more	1.06	[0.46–2.43]	0.89
Any packaged sweets	1.25	[0.68–2.30]	0.469
Number of packaged sweets on previous day			
One	2.22	[1.11–4.44]	0.02
Two or more	0.46	[0.15–1.35]	0.16

^†^ Odds ratios were generated using multivariate logistic regression; separate models adjusted for subject assignment in the original SMILE study were run for each exposure variable (no other exposure variables met criteria for inclusion); N = 360–362 (cases with missing relevant data were excluded from analysis); ^‡^ Flavored sugary drinks include fruit juice, sodas, and syrups; foods of low nutritional value (FLNV) include packaged salty snacks, packaged sweets, and Khmer sweet cake.

## Data Availability

The data presented in this study are available upon reasonable request to the corresponding author. The data are not publicly available due to privacy concerns for participants.

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
