# Peer review of "Sugar-Sweetened Beverages, Foods of Low Nutritional Value, and Child Undernutrition in Cambodia"

_ijerph, 2024, doi:10.3390/ijerph21020169_

Round 1
Reviewer 1 Report
Comments and Suggestions for Authors
Dear authors,
Here are some recommendations for change.
Introduction section:
Authors should identify why it was conducted in rural and semi-urban and not urban areas.
This section should relate the consumption of foods of low nutritional value and health problems such as obesity, diabetes and others in Asian countries. For example: Veit M, van Asten R, Olie A, Prinz P. The role of dietary sugars, overweight, and obesity in type 2 diabetes mellitus: a narrative review. Eur J Clin Nutr. 2022;76(11):1497-1501. This will enhance the novelty and interest of this article. It should also be related to growth retardation and wasting.
In Table 1 in the results section, importance is given to Chest infection in the last month, diarrhea >3 days in the last month, and antibiotic intake in the last month. This information should be justified in the introduction or methodology.
Methodology section:
On line 80 and in the same paragraph it is not stated that the exam questionnaire or questions are self-developed. This is a limitation that should be indicated in the discussion section.
The authors should provide supplementary file S1.
It should show that the concepts of worsening or declining health have been validated in other research and are self-developed should be indicated in this section and indicate in the discussion that it is a limiting factor.
Results section:
Tables should be self-explanatory, so the meaning of abbreviations should be indicated, therefore, Table 1, 2, 3 and 4 should be revised.
In each of these sections (3.1, 3.2, 3.3, 3.4 and 3.5) a short summary should be included at the end indicating the relationship between bad habits and worsening health.
Discussion section
It is recommended to be divided by age comments and health problems.
Best regards.
Reviewer 2 Report
Comments and Suggestions for Authors
Many researches about sugar-sweetened beverages and foods of low nutritional value with the risk of being obese, but little is known about the relationship with undernutrition.maybe it's interesting, but some problems be resolved.
1.more information about why sugar-sweetened beverages and foods of low nutritional value lead to undernutriton should be given out in background.
2.more information about dietary should be given out, and maybe be taken as adjusted factors.
3.more detailed information about physical measurement and survey should be given out.
4.more information about why and how sugar-sweetened beverages and foods of low nutritional value lead to undernutriton should be discusssed in discussion part.
Reviewer 3 Report
Comments and Suggestions for Authors
Suggestions for Authors
Here are my comments and suggestions to improve the quality of your manuscript:
1-Line 86, Water was included in the nutritious beverages category, but it does not contribute to the caloric and/or nutrient intake. Specify this point and increase this part in your sentences.
2-Lines 90-91 ". A “packaged food” variable was created by combining packaged salty snacks and packaged sweets." – Please give more details on this concept: what kind of package sweets or salty snakes were included ( e.g. chocolate, cakes, candies, etc.) and what was the nutrient composition of this package.
3-Lines 23-24 – it was mentioned the registration of food intake from the previous day. Please be more specific in the methodology on the type of nutritional evaluation utilized in the study (e.g., 24 HR). In the table 2 (line 175_ the number of portions was provided and used as a predictor. If the nutrient intake evaluation was not used in the study, please mention this point and be more explicit in the methodology on the questionnaire used to collect the food intake.
4-Lines 78-79 - “Data were collected from 2015 to 2017 by trained primary healthcare providers.” – The study addresses a period of two years from 5 years ago—explain the relevance of this period for the actual moment. Is the food intake registered in the study still relevant to 2023?
5-Lines 125-127 Has the frequency of malnutrition remained unchanged during the last 5 years? Are these results still relevant? Please give more details in the discussion section.
6-Lines 134-136 – “By 6 months of age, a larger proportion of infants had been introduced to a wider 134 variety of foods and beverages. Roughly a third to less than half of the children had been 135 introduced to nutrient-dense foods including fruit, meat, egg, and vegetables.” Provide a detailed table to depict in a more precise manner these results.
Round 2
Reviewer 1 Report
Comments and Suggestions for Authors
Dear authors, thank you for your responses and I comment on my latest recommendations for change.
Methodology section:
In the paragraph on line 79 it is stated that the dietary questionnaire came from the Cambogia SMILE study which corresponds to the corresponding reference (Evaluation of a community-based early 367 childhood caries (ECC) intervention in Cambodia. Community Dent. Oral Epidemiol. 2021, 49, 275-283). However, it seems that this questionnaire was not designed to analyze the dietary habits of children in terms of energy intake and macronutrients ingested. If this is the case, this questionnaire should be validated against another tool that has been validated and the level of precision of the data observed should be observed, and I give two examples:
Saravia L, González-Zapata LI, Rendo-Urteaga T, Ramos J, Collese TS, Bove I, et al. Development of a Food Frequency Questionnaire for Assessing Dietary Intake in Children and Adolescents in South America. Obesity (Silver Spring). 2018;26 Suppl 1:S31-S40.
Sotos-Prieto M, Moreno-Franco B, Ordovás JM, León M, Casasnovas JA, Peñalvo JL. Design and development of an instrument to measure overall lifestyle habits for epidemiological research: the Mediterranean Lifestyle (MEDLIFE) index. Public Health Nutr. 2015 Apr;18(6):959-67.
My advice is to look at already validated questionnaires and adapt them to your population.
Based on the background commented, if the authors cannot demonstrate otherwise, the discussion should indicate that in the future they should work on the validation of a questionnaire on the intake of various components: foods of low nutritional value, sugar-sweetened beverages, among others. This comment should be complemented in the conclusions section where it should be indicated that this is a very topical subject and there are still areas to be investigated.
Best regards.
Reviewer 2 Report
Comments and Suggestions for Authors
no
Author Response
Thank you for your review.
Reviewer 3 Report
Comments and Suggestions for Authors
The manuscript has undergone significant enhancements and I have no other remarks to contribute.
Author Response
Thank you for your review.